# Effect of a Trace Addition of Sn on the Aging Behavior of Al–Mg–Si Alloy with a Different Mg/Si Ratio

**DOI:** 10.3390/ma13040913

**Published:** 2020-02-19

**Authors:** Lehang Ma, Jianguo Tang, Wenbin Tu, Lingying Ye, Haichun Jiang, Xin Zhan, Jiuhui Zhao

**Affiliations:** 1Light Alloy Research Institute, Central South University, Changsha 410083, China; 173812047@csu.edu.cn (L.M.); lingyingye@csu.edu.cn (L.Y.); zjh_aluminium@csu.edu.cn (J.Z.); 2School of Materials Science and Engineering, Central South University, Changsha 410083, China; tuwenbin225@163.com (W.T.); 183101031@csu.edu.cn (X.Z.); 3MINTH Research and Development Center, Anji 313300, China

**Keywords:** Al–Mg–Si alloy, Mg/Si ratio, trace Sn, artificial aging, activation energy

## Abstract

In this paper, the effect of trace Sn on the precipitation behavior and mechanical properties of Al–Mg–Si alloys with different Mg/Si ratios aged at 180 °C was investigated using hardness measurements, a room-temperature tensile test, transmission electron microscopy and differential scanning calorimetry. The results shown that Sn reduces the precipitation activation energy, increases the number density of β″ precipitates, and then increased the aging hardenability and mechanical properties of the Al–Mg–Si alloy. However, the positive effect of Sn on the mechanical properties of the Al–Mg–Si alloy drops with the decrease of the Mg/Si ratio of the alloy.

## 1. Introduction

The Al–Mg–Si alloy is widely used in the production of automobile body panels for lightweight passenger cars due to its good formability, corrosion resistance and medium strength [1,2,3]. This kind of alloy can be strengthened by a series of heat treatments, including solution treatment, quenching and then artificial aging, in which a high number density of nanometer-sized strengthening precipitates are developed [4,5,6]. In this series of alloys, the widely accepted precipitation sequence is as follows: supersaturated solid solution (SSSS) → atomic cluster → GP zone → pre β″/ β″ → β′ → β [7,8,9,10], in which the β and β″ is the stable phase, and β″ is considered to be the most effective strengthening precipitates of the Al–Mg–Si alloy, the stoichiometric composition of which is supposed to be Mg_5_Si_6_ [9,10] or Mg_4_Al_3_Si_4_ [11]. Therefore, the underlying mechanism for strengthening this series of alloys is to accelerate and increase the number density of the β″ precipitates.

In the commercial production process of an Al–Mg–Si alloy sheet for automobile body panels, the alloy sheets are not artificially aged immediately after quenching, but are kept at room temperature for a period of time. During this period, natural aging would take place, which will decrease the formability of the alloy and adversely affect the subsequent artificial aging effect [12,13,14]. To reduce the adverse impact of the natural aging effects, methods such as pre-aging, pre-straining, and interrupted quenching of the alloy, are widely used [15,16,17,18].

In recent years, several investigations [19,20,21,22,23,24,25] have revealed that the trace addition of Sn in this Al–Mg–Si alloy can reduce the negative effect of natural aging. At present, it is generally accepted that Sn can reduce the number density of atomic clusters formed in the alloys during the natural aging process [19,20,21,25], and thereby suppress the natural aging effect of the alloy.

Tu et al. found that the effect of Sn increased along with the amount added [21]. However, there are varying viewpoints on the effect of Sn on the artificial aging of the Al–Mg–Si alloy. Shishido et al. and Cheng et al. found that Sn can not only prevent the negative effect of natural aging, but also adversely affects subsequent artificial aging at 170 °C [19,20]. However, Liu et al. and Xiang et al. found that the precipitation sequence of alloy with high Mg content aged at 250 °C could be changed by adding Sn, and therefore the age-hardening ability of this alloy can be significantly improved [23,24]. Zhang et al. found that the age-hardening ability of 6014 alloy aged at 180 °C and higher was increased by adding Sn [25]. Tu et al. found that the adverse effect of Sn on the artificial aging behavior of natural aged Al–Mg–Si alloy could be overcome by an increased artificial aging temperature [21] or joint addition of Sn and Cu [22].

Werinos et al. found that the natural and artificial aging of the Sn-containing Al–Mg–Si alloy were substantially affected by the concentration of Mg, Si and Sn [26]. However, what role which the ratio of Mg/Si would play on the effect of Sn is unclear at present. Therefore, the effect of adding 0.1% wt. of Sn on the artificial aging behavior of alloy with three different Mg/Si ratios aged at 180 °C was investigated in the present paper, which provides experimental data and literature references for the design of an Al–Mg–Si alloy.

## 2. Experimental

The investigated alloys with total Mg + Si mass faction of 1.6%, but three different ratios of Mg/Si, are shown in Table 1. 0.1% wt. Sn was added to the three investigated alloys with different Mg/Si ratios, which were shown in Table 1. The Mg/Si ratio of the H–M and H–M–Sn alloy were 1.68 and 1.69 respectively, which were close to that of the equilibrium precipitates β; the Mg/Si ratio of the B–M and B–M–Sn alloy were 1.01 and 0.99, respectively, which were close to that of the main strengthening precipitates β″; the Mg/Si ratio of the L–M and L-M–Sn alloy were 0.58, which means that there was excessive Si according to the β″precipitates. The amount of Sn added is set 0.1% according to the investigation of Tu et al [21].

Pure aluminum, pure magnesium and master alloys (Al-30% wt. Si, Al-10% wt. Sn and Al-10% wt. Ti) were melted in a graphite crucible with a resistance furnace (SG 2-7.5-10XP, ZSST, Dongtai, JS, China) at 750 °C. The molten metals were casted into a copper mold after being degassed with C_2_Cl_6_ at 720 °C. The as-cast ingots were homogenized at 540 °C for 30 h. After being scalped to remove the surface layer, the ingots were hot rolled from 42 mm to 7 mm at 480 °C. The hot rolled slabs were annealed at 420 °C for 2 h and then cold-rolled down to a thickness of 1 mm. Solution heat treatment of the cold-rolled sheets were performed at 570 °C for 30 min, followed by water quenching to room temperature. An air circulating furnace (SGMA Z4/10A, SIGMA, Luoyang, HN, China) was used for the above-mentioned manufacture process.

Hardness was tested by the HV-10B Vickers hardness tester with a load of 4.9 N and indention time of 10 s. The hardness of the alloy was the average value of five indentations. The tensile test was carried out on the MTS810 computer-controlled test machine (MTS, Faribault, MN, USA) with a load speed of 2 mm/min at room temperature. The stretching direction was along the rolling direction of the sheets. The tensile properties were the mean value of three parallel specimens.

The DSC curves of the as-quenched specimens were tested by the DSC-700 instrument (METTLERTOLED, Shanghai, China). The weight of specimen for DSC test is 10 mg. The specimens were heated from room temperature to 500 °C with the heating rate of 10 °C/min.

The reference curve of pure Al, which was measured using the same heating schedule, was used as the baseline and subtracted from the heat flow curves of all of the tested specimens.

The FEI Tecnai G2 F20 transmission electron microscope (FEI, Hillsboro, OR, USA) operated at 200 kV was used to investigate the precipitates in the investigated alloys. The TEM specimens were mechanically polished to the thickness of about 100 μm and punched into discs with the diameter of 3 mm. The discs were polished with electrolyte consisting of 30% nitric acid in methanol down to appropriate thickness by the Twin-jet polish instrument. The polish was operated at temperature from −35 °C to −25 °C. The length, diameter and density of the precipitates in the TEM image were counted using Image-pro plus, and the statistical results of the precipitates were obtained from about 20 TEM images of each specimen.

## 3. Results

### 3.1. Evolution of Hardness during Artificial Aging

The evolution of hardness of the six investigated alloys during artificial aging at 180 °C were illustrated in Figure 1. The hardness of the H–M alloy increased rapidly from 40 HV to 70.8 HV, and the H–M–Sn alloy increased from 36.5 HV to 55.1 HV after artificial aging at 180 °C for 10 min. But the aging hardening rate of the H–M–Sn alloy is higher than that of the H–M alloy during the subsequent artificial aging after that. The time required to reach peak hardness for the H–M–Sn and H–M alloys were 8 and 12 h, respectively, and the hardness of the peak aged were 126 and 117.5 HV, as shown in Figure 1a. The hardness of the B–M alloy increased from 42 HV to 75.8 HV and the B–M–Sn alloy increased from 39.8 HV to 52 HV after artificial ageing at 180 °C for 10 min. Then the aging hardness rate of this same B–M–Sn alloy is higher than that of the B–M alloy during artificial aging at 180 °C from 10 to 70 min. The hardness of both the B–M and B–M–Sn alloys were 98.4 HV after artificial aging at 180 °C for 70 min, and no obvious effect of Sn on the artificial aging of B–M alloys was found in the subsequent artificial aging at 180 °C. The times required to reach peak hardness for both B–M and B–M–Sn alloys were 8 h, and their peak hardness of them was 128 HV, as shown in Figure 1b. Figure 1c illustrates the hardness evolution of the alloys with a 0.58 Mg/Si ratio aged at 180 °C. L–M, and the aging hardening rate of the L-M–Sn alloys, were almost the same when they were aged at 180 °C for no more than 90 min. But the aging hardness rate of the L-M–Sn alloy is higher than the L–M alloy by further increasing the aging time. The times required to reach peak hardness for the L-M–Sn and L–M alloys were 3 and 5 h, respectively, whose peak hardness values were 116 and 119.6 HV, as shown in Figure 1c. According to the investigation of Tu et al. [21] and Liu et al. [27] the formation of atom clusters can be inhibited by the addition of Sn. The initial stage of artificial aging, which is also involves the formation of atom clusters, could also be suppressed by the addition of Sn, and therefore the hardness of the Sn-added alloys is lower than that of corresponding Sn-free alloys when they were artificially aged at 180 °C for 10 min. During the subsequent artificial aging, the β″ precipitates were precipitated in the Al matrix, thereby the hardness of the alloy. The density of β″ precipitates increased by addition of Sn, and the aging hardening rate of the Sn-added alloys is higher than that of corresponding Sn-free alloys during the subsequent artificial aging.

### 3.2. Tensile Results

Effects of the added of Sn on the mechanical properties of peak aged Al–Mg–Si alloys with different Mg/Si ratios were further investigated through tensile tests at room temperature, which was shown in Figure 2, and the detailed mechanical properties of the six peak aged experimental alloys were shown in Figure 2d and Table 2. For the alloys with 1.68 Mg/Si ratio, the yield strength and ultimate tensile strength increased by 22.7 and 17.2 MPa by the addition of Sn. In the alloys with the 1.00 Mg/Si ratio, the yield strength and ultimate tensile strength increased by 4.8 and 5.1 MPa by the addition of Sn. But in the alloys with 0.58 Mg/Si ratio, the yield strength and ultimate tensile strength decreased by 12.5 and 5.8 MPa by the addition of Sn. Those results of tensile tests were in agreement with the hardness results shown in Figure 1. The mechanical properties of the peak aged alloys with 1.68 Mg/Si ratio and 1.00 Mg/Si ratio were increased by the addition of Sn. But the mechanical properties of the peak aged alloys with 0.58 Mg/Si ratio were decreased by the addition of Sn.

### 3.3. Microstructure Investigation

Figure 3 showed the TEM images and the selected area diffraction pattern (SADP) of the experimental alloys peak aged at 180 °C. All the TEM images were acquired along <001>_Al_ zone axes. A large number of uniformly distributed dot-like and need-like precipitates were found in the TEM images of the six investigated alloys. The Cross-shaped diffraction pattern near to that of (110)_Al_ matrix were found in the SADPs. It can be assumed from the shape of precipitates and corresponding SADP that the precipitates were β″, which are considered to be the main strengthening precipitates in Al–Mg–Si alloys. In the peak aged alloys, the length and diameter of the β″ precipitates decreased, and the density of the β″ precipitates increased with the decrease of the Mg/Si ratio. The number density of precipitates in the alloys containing Sn was higher than that in the corresponding alloys without any addition of Sn, and the length and diameter of precipitates in the alloys containing Sn were smaller than those in the alloys without the addition of Sn, which were shown in Figure 3. In addition, the length and diameter distribution of β″ precipitates in the peak-aged alloys are displayed in Figure 4 and Figure 5, respectively.

The average length of β″ precipitates increased with the decrease of Mg/Si ratio in the Sn-free alloys, but the average length of β″ precipitates decreased with the decrease of the Mg/Si ratio in the Sn-added alloys. The average diameter of β″ precipitates decreased with the decrease of the Mg/Si ratio in both the Sn-free alloys and Sn-added alloys. In order to quantitatively analyze the size and distribution of β″ precipitates in the experimental alloys, the statistical results calculated from Figure 3, Figure 4 and Figure 5 are list in Table 3. It could be found that with the decrease of the Mg/Si ratio, the average length and density of β″ precipitates increased from 17.3 nm and 2.55 × 10^15^/m^2^ (H–M alloy) to 23.1 nm and 3.05 × 10^15^/m^2^ (L–M alloy), and the average diameter of β″ precipitates decreased from 3.9 nm (H–M alloy) to 3.3 nm (L–M alloy) in Sn-free alloys. In Sn-added alloys, the average diameter of the β″ precipitates decreased from 3.6 nm (H–M–Sn alloy) to 2.7 nm (L-M–Sn alloy) and the average density of the β″ precipitates increased from 3.55 × 10^15^/m^2^ (H–M–Sn alloy) to 4.30 × 10^15^/m^2^ (L-M–Sn alloy) with the decrease of Mg/Si ratio. The average length of the β″ precipitates in H–M–Sn and B–M–Sn alloys was at the same level, while a sharp decrease was found in the L-M–Sn alloy. Besides, compared with the Sn-free alloys, the Sn-added alloys showed a significant decrease in the average length of the β″ precipitates, and a slight decrease in the average diameter of the β″ precipitates, along with an obvious increase in average density of the β″ precipitates. The length and diameter of the β″ precipitates of alloys with 1.00 and 0.58 Mg/Si ratios decreased substantially by the addition of Sn, and the smaller β″ precipitates have a positive effect on the elongation rate of the alloy. The elongation rate of this alloy with the 1.00 and 0.58 Mg/Si ratios was increased by the addition of Sn, as shown in Figure 2d and Table 2. However, the density of β″ precipitates increased significantly without a not significant decrease of the length and diameter of the β″ precipitates in the alloy with 1.68 Mg/Si ratio, that have a negative effect on the elongation ratio of the alloy. The elongation ratio of the alloy with 1.68 Mg/Si ratio was decreased by the addition of Sn, as shown in Figure 2d and Table 2.

### 3.4. DSC Analysis

The results of the differential scanning calorimeter (DSC) results of the six as-quenched alloys are shown in Figure 6a. Three different exothermic peaks corresponding to different precipitations, I to β″, II to β′ and III to β, were found in all DSC curves of the investigated alloys. As shown in Figure 6b, the peak temperature for the precipitation of β″ precipitates were decreased about 10 °C by the addition of Sn, but the peak temperature for the precipitation of β″ precipitates was not significantly affected by the Mg/Si ratio.

In order to investigate the effect of Sn and the Mg/Si ratio on the thermodynamic behavior of β″ precipitates quantitatively, the Gaussian model function is used to separate the exothermic peak of β″ precipitates from the DSC curves of the investigated alloys, which are shown in Figure 7a. The Gaussian model results in Figure 7a show that no significant effects of the Mg/Si ratio on the peak temperature of β″precipitates is found. It is found that only the peak temperature of β″ precipitates in our L–M alloy is approximately 6.5 °C lower than that in H–M and B–M alloys. But the peak temperature of the β″ precipitates is significantly affected by the addition of Sn. The addition of Sn decreases the peak temperature of β″precipitates in this H–M alloy from 249.6 °C to 243.9 °C in the H–M–Sn alloy, and the decrement is 5.7 °C. The addition of Sn decreases the peak temperature of β″precipitates in the B–M alloy from 252.8 °C to 242.3 °C in the B–M–Sn alloy, and the decrement is 10.5 °C. Additionally, the addition of Sn decreased the peak temperature of β″precipitates in the L–M alloy from 244.7 °C to 236.7 °C in the L-M–Sn alloy, and the decrement is 8 °C. While reducing the peak temperature of β″ precipitates, Sn can also increase the precipitation rate of the alloy at β″ precipitates.

The area of the separated Gaussian model peak corresponds to the volume fraction of β″ precipitates. Therefore, the separated Gaussian model functions are used to calculate the transformation rate of the β″ precipitates during the heating process; i.e.,
(1)$$Y\left(T\right)=A_{T}/A_{f}$$
where *A_T_* is the integrated area of the part of the Gaussian model function terminated at temperature *T*, and *A_f_* is the total area of the complete separated Gaussian model function, and the evolution of the transformation rate during the heating process of the six investigated alloys is shown in Figure 7b. The transformation rate of β″ precipitates were increased by the addition of Sn. The transformation rate of β″ precipitates were not significantly affected by the Mg/Si ratio in the Sn-free alloys. But the transformation rate of β″ precipitates were increased in the Sn-containing alloys with the decrease of the Mg/Si ratio.

The evolution of the volume fraction of β″ precipitates could also be described by the Avrami-Johnson–Mehl–Avrami equation [28,29,30]
(2)$$Y=1-\exp\left[-{\left(kt\right)}^{n}\right]$$
(3)$$k=k_{1}\exp\left(-\frac{Q}{RT}\right)$$
where *t* is the time for precipitation, *k*_1_ and *n* are material constants, *Q* is the activation energy for precipitation of β″, R is the gas constant of 8.314 J/(mol·k) and *T* is temperature. Difference in parameters k and n represent difference in the nucleation and growth behavior, and a value of 1.5 was found to be the optimal value of the Avrami index (*n*) for a description of the DSC curves analyzed [22,31].

The transformation rate of β″ precipitates can be calculated according to Equation (4), i.e.,
(4)$$\frac{dY}{dt}=kf\left(Y\right)$$
(5)$$f\left(Y\right)=n\left(1-Y\right){\left[1-\ln\left(1-Y\right)\right]}^{\left(n-1\right)/n}$$
and the evolution of the transformation rate with time t can be expressed as follows:(6)$$\;\;\frac{dY}{dt}=\left(\frac{dY}{dT}\right)\left(\frac{dT}{dt}\right)=\Phi *\left(\frac{dY}{dT}\right)$$
where Φ = *dT*/*dt* is the heating rate, and it is 10 °C/min in the DSC test of the present paper. Now, *dY*/*dT* can be calculated from Figure 7b, and the result is shown in Figure 7c. As shown in Figure 7b,c, the volume fraction *Y* and precipitation rate of β″ precipitates in the Sn-added alloys are higher than those in the Sn-free alloys within the temperature range of 200 °C to 240 °C.

The following equation can be derived from Equations (3), (4) and (6), i.e.,
(7)$$\;\;\;\;\;\;\;\;ln\left[\left(\frac{dY}{dT}\right)\left(\frac{\Phi }{f\left(Y\right)}\right)\right]=lnk_{0}-\left(-\frac{Q}{R}\right)\left(\frac{1}{T}\right)$$

The above equation means that there is a linear relationship between *ln*[(*dY*/*dT*) * Φ/*f*(*Y*)] and 1/*T*. By assuming n = 3/2, *f*(*Y*) can be calculated from the result shown in Figure 7b and Equation (5). Therefore, *ln*[(*dY*/*dT*) * Φ/*f*(*Y*)] at specific temperature can be calculated, and by a linear fitting of *ln*[(*dY*/*dT*) * Φ/*f*(*Y*)] and 1/*T*, the activation energy for precipitation of β″ can be calculated according to Equation (7).

The fitting results are shown in Figure 7d and Table 4. The activation energy of β″ precipitates in H–M and B–M alloys are 74.06 kJ/mol and 74.65 kJ/mol, respectively, but that in the L–M alloy, it is 67.84 kJ/mol, which is lower than that of the H–M and B–M alloys. The activation energy of β″ precipitates can be significantly reduced by the addition of Sn, the activation of β″ precipitates in the H–M–Sn alloy, B–M–Sn alloy and L-M–Sn alloy are 44.58 kJ/mol, 41.32 kJ/mol and 47.43 kJ/mol, respectively, which is 29.48 kJ/mol, 33.33 kJ/mol and 20.41 kJ/mol lower than that in corresponding alloys without the addition of Sn.

The effect of Sn on the β″ precipitates of the Al–Mg–Si alloy during artificial aging is then analyzed from the thermodynamic perspective. The results of the DSC experiment are shown in Figure 6b: the addition of Sn decreases the starting precipitation temperature of β″ precipitates from 218 °C to 200 °C, and decreases the peak precipitation temperature of β″ precipitates by approximately 8 °C. The calculated results of the activation energy Q of β″ precipitates are listed in Table 4. The average activation energy of β″ precipitates decreases by 27.8 kJ/mol by the addition of Sn, which promotes the precipitation of β″ precipitates, and an increased density of β″ precipitates. The decrease of the activation energy of the β″ precipitates can be mainly attributed to Sn that affects the precipitation sequence of the alloy, and the schematic illustrations, as shown in Figure 8. Sn captures vacancies and prevents the formation of atomic clusters in the matrix during the initial stage of artificial aging [18,19,20]. This reason caused that the precipitation model of the β″ precipitates have changed, it directly precipitates the β″ precipitates by a supersaturated solid solution. This change in the precipitation mode reduces the activation energy of the β″ precipitates. Liu and Xiang’s research the similar result that Sn changes the precipitation sequence of the alloy when the Al–Mg–Si alloys were aged at 250 °C after quenching; the β″ precipitates are directly precipitated from the supersaturated solid solution in the matrix [23,24].

## 4. Discussion

It is well known that the formation of atom clusters is involved in the initial stage of the artificial aging of Al–Mg–Si alloys before large amount β″ precipitates are formed [19,20,21,22]. But the evolution of these clusters differs: (I) remain stable, dissolved and them nucleation of β″ precipitates and (II) nucleation or transformation of β″ precipitates from these clusters [8,9,10] according to the chemical composition and size of the clusters.

As the concentration of Mg in the H–M alloy is high, the Mg-rich clusters with atomic ratio of Mg/Si more than 1.4 [32], which were also found by Tu et al. [21], can be easily formed during initial stage of artificial aging. While the formation of clusters would be suppressed by the addition of Sn as reported by Tu et al. [22], therefore the hardness of the H–M alloy is higher than that of the H–M–Sn alloy, as shown in Figure 1. These Mg-rich clusters are assumed to be unstable according to the investigation of Aruga et al. [32], and therefore on the substantial endothermic peak before the exothermic peak of the β″ phase is found in the DSC curves of the H–M alloy, as shown in Figure 6. As the atom clusters formed in the H–M–Sn alloy are fewer than that in the H–M alloy, fewer atom clusters means a higher concentration of solute atoms and faster precipitation of β″ precipitates, as shown in Figure 3, Figure 4 and Figure 5, and therefore the increase of hardness of the H–M–Sn alloy is higher than that of the H–M alloy during the subsequent artificial aging, as shown in Figure 1a and Figure 2a.

Large numbers of atom clusters with an atom ratio of Mg/Si close to 1 are assumed to be developed in alloys with 1.00 Mg/Si ratio (B–M) during the initial stage of artificial aging according to several investigations [11,33,34]. The suppression of the formation of clusters by the addition of Sn is assumed to be the reason for the lower hardness of the B–M–Sn alloy than the B–M alloy during the initial stage of artificial aging, as shown in Figure 1b. As the atomic Mg/Si ratio of clusters formed in the B–M and B–M–Sn is close to 1, they are supposed to be able to be transformed into β″ precipitates easily during subsequent artificial aging according to the investigation of Ninive et al. [11]. Therefore, no significant effect of Sn on artificial aging is found, as shown in Figure 1b and Figure 2b, and the peak hardness of the B–M alloy is even a little high than that of the B–M–Sn alloy.

Large numbers of Si-rich clusters with atomic Mg/Si ratios less than 2/3 would be formed during the initial stage aging of the L–M alloy according to previous investigations [32,35]. The size of these Si-rich clusters is about 10 solute atoms, which is smaller than other clusters, and therefore the increase of hardness of the L–M alloy is smaller than that of the H–M and B–M alloys during the initial stage of artificial aging. As the diffusion of Si in Al is higher than that of Mg, the excess Si in the L–M alloy would accelerate the precipitation, and therefore the time required for peak aging of our L–M alloy is shorter than that of the alloys with the 1.68 and 1.00 Mg/Si ratios, as shown in Figure 1. The Si-rich clusters are assumed to be stable, which means that they cannot be dissolved nor transformed into β″ precipitates by subsequent artificial aging at 180 °C, according to Serizawa et al. [34]. As the development of such stable Si-rich clusters is suppressed by the addition of Sn, the concentration of solute atoms in the L-M–Sn alloy would be higher than that in the L–M alloy, which means a higher nucleation rate during subsequent aging. Therefore, the number density of β″ precipitates in the L-M–Sn alloy is higher than that in the L–M alloy, as shown in Figure 3, Figure 4 and Figure 5. However, the addition of Sn significantly decreased the average length and diameter of the β″ precipitates by 14.2 nm and 0.5 nm, respectively, in alloys with a 0.58 Mg/Si ratio, as shown in Figure 3e,f and Table 3, and this weakens the strengthening effect of these β″precipitates. Therefore, the addition of Sn decreased the mechanical properties of the alloy with the 0.58 Mg/Si ratio at peak aged, as shown in Figure 1c and Figure 2c.

## 5. Conclusions

Herein, the effect of Sn on the artificial aging behavior of Al–Mg–Si alloys with three different Mg/Si ratios at 180 °C was investigated, and it was found that Sn can reduce the activation energy of β″ precipitates, promote the precipitation of β″ precipitates, and improve the age-hardening ability of the alloy. However, the effect of Sn on the mechanical properties of the peak aged alloy drops with the decrease of the Mg/Si ratio of the alloy. The main conclusions be summarized as follows:The addition of 0.10% wt. Sn to alloys with a 1.68 Mg/Si ratio can improve the age-hardening ability of the alloy during artificial aging at 180 °C. Sn reduces the activation energy of the β″ precipitates, increases the density of the β″ precipitates, and significantly increases the yield and tensile strength of the peak aged alloy by 22.7 and 17.2 MPa, respectively.The addition of 0.11% wt. Sn to alloys with 1.00 Mg/Si ratio can reduce the aging hardenability of the alloy during artificial aging at 180 °C, but no significant effect on the hardness of the alloy after peak aging can be found. Sn reduces the activation energy of the β″ precipitates, increases the density of the β″ precipitates, and increases the yield and tensile strength of the peak aged alloys by 4.8 and 5.1 MPa, respectively.The addition of 0.11% wt. Sn to alloys with the 0.58 Mg/Si ratio can improve the age-hardening ability of the alloy during artificial aging at 180 °C, but can reduce the peak hardness of the alloy. In alloys with 0.58 Mg/Si ratio, although Sn can reduce the activation energy of the β″ precipitates and increase the density of the β″ precipitate, it also prevents the growth of the β″ precipitate, reducing the length and diameter of the β″ precipitate. Therefore, Sn weakened the strengthening effect of β″ precipitates in the alloy with 0.58 Mg/Si ratio, and decreased the yield and tensile strength of the peak aged alloy by 12.5 and 5.8 MPa, respectively.

## Figures and Tables

**Figure 1 materials-13-00913-f001:**
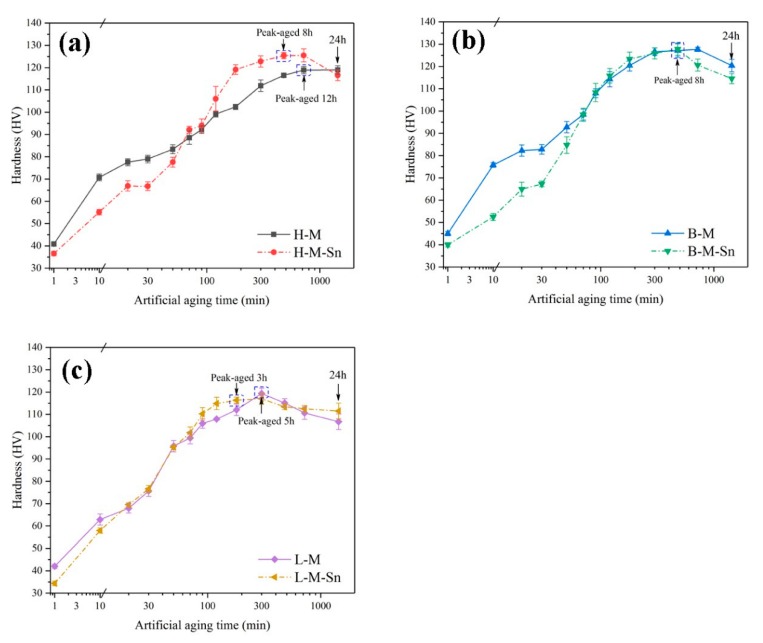
Evolution of hardness during artificial ageing at 180 °C for the investigated alloys: (**a**) H–M and H–M–Sn alloys; (**b**) B–M and B–M–Sn alloys; (**c**) L–M and L-M–Sn alloys.

**Figure 2 materials-13-00913-f002:**
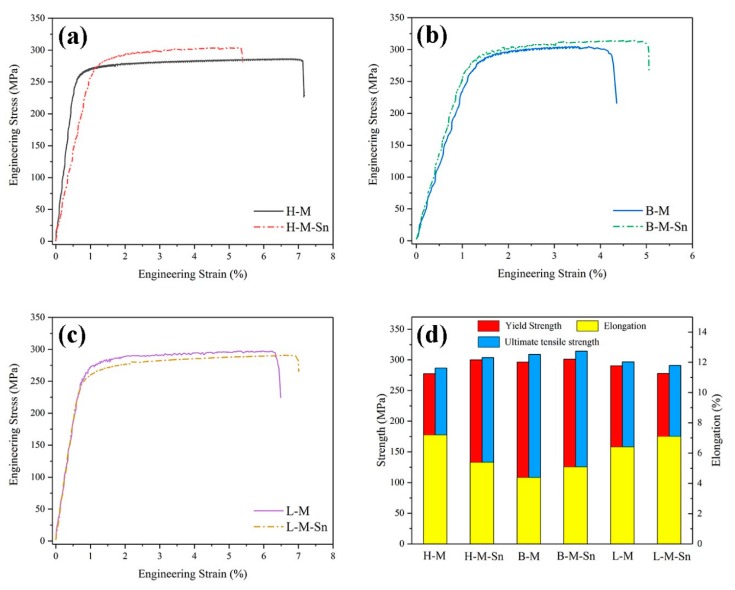
Room temperature tensile curves and mechanical properties of the peak aged: (**a**) H–M and H–M–Sn alloys; (**b**) B–M and B–M–Sn alloys; (**c**) L–M and L-M–Sn alloys; (**d**) mechanical properties.

**Figure 3 materials-13-00913-f003:**
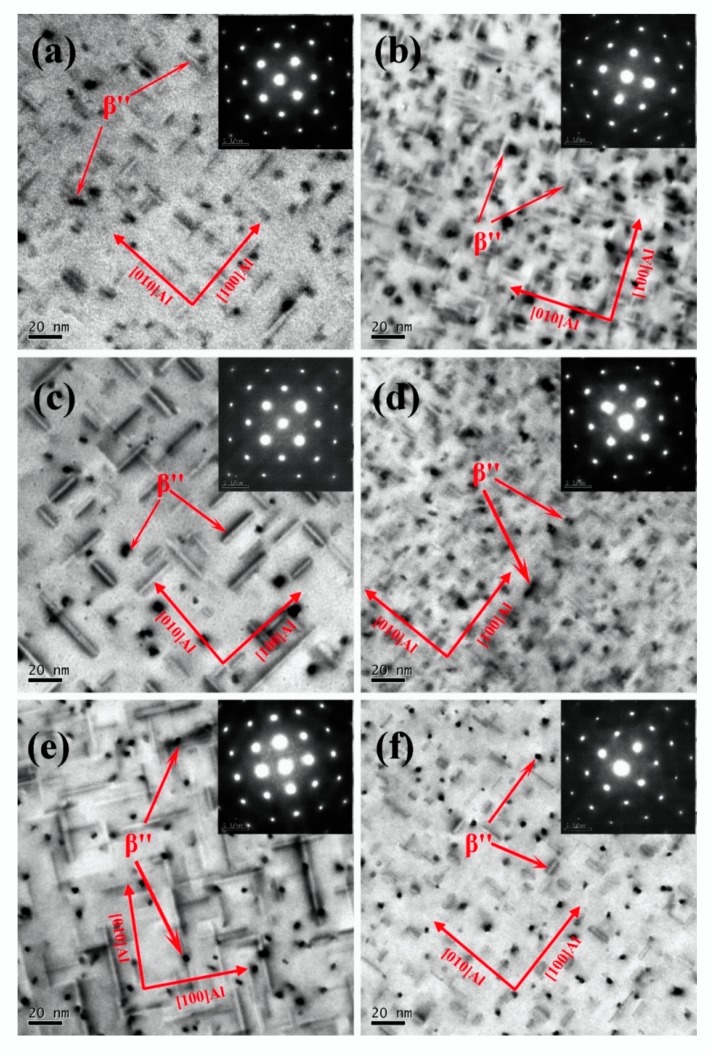
Transmission electron microscopy (TEM) images of the six investigated alloys artificially aged to peak hardness: (**a**) H–M; (**b**) H–M–Sn; (**c**) B–M; (**d**) B–M–Sn; (**e**) L–M; (**f**) L-M–Sn.

**Figure 4 materials-13-00913-f004:**
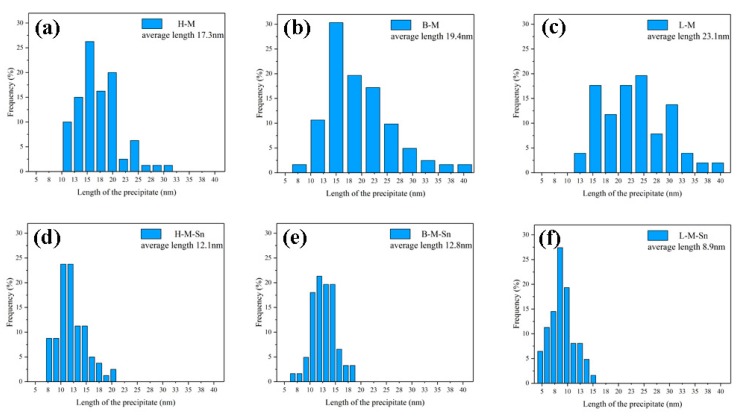
Average length and length distribution of the β″ precipitates in the peak aged alloys: (**a**) H–M; (**b**) B–M; (**c**) L–M; (**d**) H–M–Sn; (**e**) B–M–Sn; (**f**) L-M–Sn.

**Figure 5 materials-13-00913-f005:**
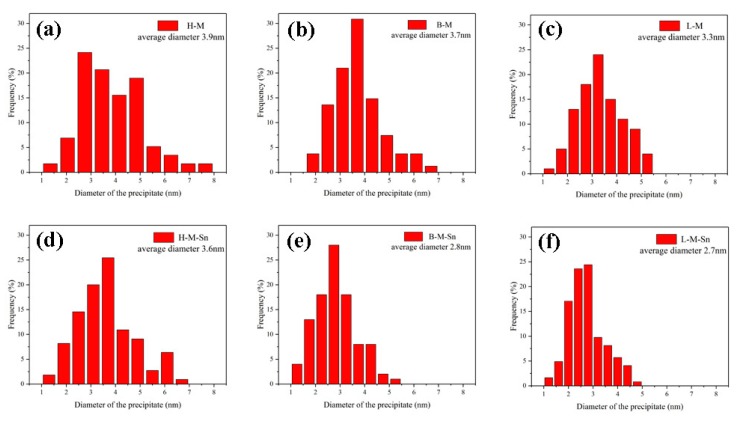
Average diameter and diameter distribution of the β″ precipitates in the peak aged alloys: (**a**) H–M; (**b**) B–M; (**c**) L–M; (**d**) H–M–Sn; (**e**) B–M–Sn; (**f**) L-M–Sn.

**Figure 6 materials-13-00913-f006:**
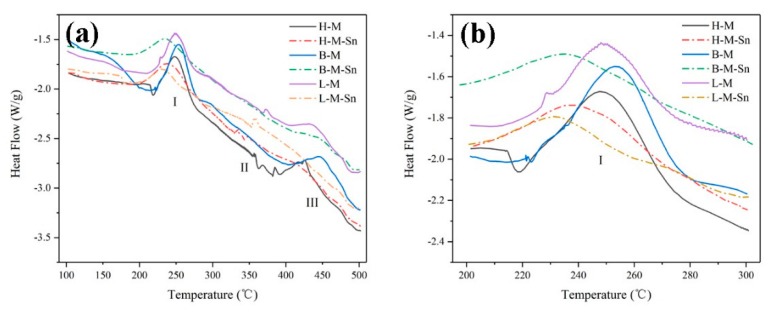
Differential scanning calorimeter (DSC) heat flow curves of six investigated alloys: (**a**) as-quenched, (**b**) precipitation peak of β″ precipitates.

**Figure 7 materials-13-00913-f007:**
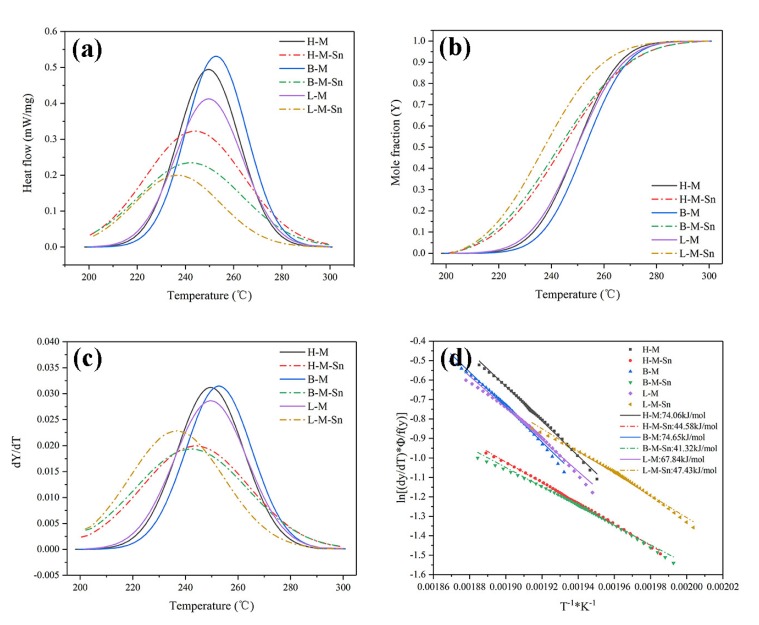
Calculation of precipitation activation energy of β″ precipitates of as-quenched alloys: (**a**) precipitation peak of β″ precipitates, (**b**) Y-T, (**c**) (*dY*/*dT*) − *T*, (**d**) *ln*[(*dY*/*dT*) * Φ/*f*(*Y*)] − 1/*T*.

**Figure 8 materials-13-00913-f008:**
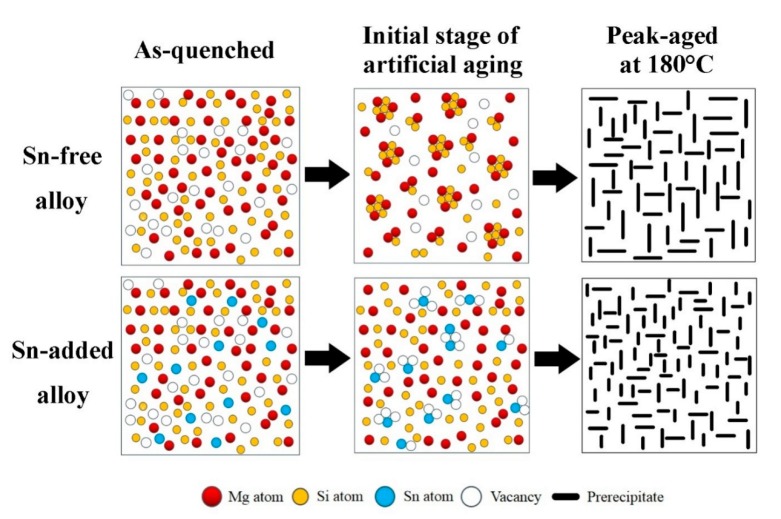
The schematic illustrations of cluster and precipitate evolution in the Sn-free and Sn-added alloys during the artificial aging at 180 °C.

**Table 1 materials-13-00913-t001:** Chemical composition of the investigated alloys (wt %).

Alloys	Mg	Si	Sn	Fe	Ti	Mg/Si Ratio
**H–M**	1.01	0.60	0.00	0.016	0.075	1.68
**H–M–Sn**	1.05	0.62	0.10	0.013	0.081	1.69
**B–M**	0.81	0.80	0.00	0.017	0.084	1.01
**B–M–Sn**	0.80	0.81	0.11	0.015	0.085	0.99
**L–M**	0.60	1.03	0.00	0.018	0.080	0.58
**L-M–Sn**	0.59	1.01	0.11	0.025	0.080	0.58

**Table 2 materials-13-00913-t002:** Tensile results of the six peak aged investigated alloys.

Alloys	H–M	H–M–Sn	B–M	B–M–Sn	L–M	L-M–Sn
**Yield Strength (MPa)**	277.4	300.1	296.3	301.1	290.3	277.8
**Ultimate Tensile Strength (MPa)**	286.6	303.8	309.1	314.2	296.7	290.9
**Elongation (%)**	7.2	5.4	4.4	5.1	6.4	7.1

**Table 3 materials-13-00913-t003:** Average length, diameter and density of the β″ precipitates at the peak aged.

Alloys	H-M	H-M-Sn	B-M	B-M-Sn	L-M	L-M-Sn
**Average length of the precipitated (nm)**	17.3	12.1	19.7	12.8	23.1	8.9
**Average diameter of the precipitated (nm)**	3.9	3.6	3.7	3.1	3.3	2.7
**Average density of the precipitate(×10^15^/m^2^)**	2.55	3.55	2.68	3.85	3.05	4.30

**Table 4 materials-13-00913-t004:** Precipitation peak temperature and activation energy of β″ precipitates in six investigated alloys.

Alloys	H–M	H–M–Sn	B–M	B–M–Sn	L–M	L-M–Sn
**Temperature of the β″ precipitation peak (°C)**	249.6	243.9	252.8	242.3	244.7	236.7
**Activation energy of β″precipitates (kJ/mol)**	74.06	44.58	74.65	41.32	67.84	47.43

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
