# Peer review of "Effect of a Trace Addition of Sn on the Aging Behavior of Al–Mg–Si Alloy with a Different Mg/Si Ratio"

_materials, 2020, doi:10.3390/ma13040913_

Round 1
Reviewer 1 Report
Dear Authors,
thank you for your contribution to materials. You present a detailed study on the effect of Sn on Aging of Al-Mg-Si alloys.
I have found the following issues:
1. you affiliations are missing. This made it hard for me to compare your work to your publication track.
2. I have one big concern, which affects the whole work:
The choice of your alloy composition is totally unclear to me. You state that you achieve a balance Mg/Si when both elements have the same weight fraction in the alloy. I totally disagree. Both elements have a different density, so you end up with a totally different atomic fraction. Furthermore, the equilibrium phase to be formed from Mg and Si is Mg2Si, which has a weight ratio Mg/Si of approx. 1.72, taking the atomic masses in account. The only useful balance state would be either „same number of atoms“ or „stoichiometric composition of Mg2Si“. Your definition is arbitrary. This has consequences for your whole study.
Luckily, your Mg/Si ratio of 1.68/1.69 is fairly close to the stoichiometric composition of equilibrium Mg2Si phase. The other two compositions then have two increasing Si excess conditions. This needs to be included in the interpretation of your results and discussion.
3. I am missing a direct comparison of the material properties you measured with results from other authors or commercial material (may also be from a data sheet of a company, cite it well). I propose to add a Strength/Elongation plot with your data points and add further data from other researchers or data sheets. If there is an offset of your properties in comparison to other works, this helps with the interpretation.
Since point 2 appears to be a complicated issue, I propose to resubmit after a major revision.
Author Response
Response to the reviewer’s comments
Reviewer #1:
Comment 1:
Advise 1: I have one big concern, which affects the whole work:
The choice of your alloy composition is totally unclear to me. You state that you achieve a balance Mg/Si when both elements have the same weight fraction in the alloy. I totally disagree. Both elements have a different density, so you end up with a totally different atomic fraction. Furthermore, the equilibrium phase to be formed from Mg and Si is Mg2Si, which has a weight ratio Mg/Si of approx. 1.72, taking the atomic masses in account. The only useful balance state would be either „same number of atoms “or” stoichiometric composition of Mg2Si“. Your definition is arbitrary. This has consequences for your whole study.
Luckily, your Mg/Si ratio of 1.68/1.69 is fairly close to the stoichiometric composition of equilibrium Mg2Si phase. The other two compositions then have two increasing Si excess conditions. This needs to be included in the interpretation of your results and discussion.
Response: Thank you for good suggestion. In the earlier investigations, researchers tended to design the alloy compositions with Mg/Si about 1.72, which were close to that of the equilibrium precipitates β. But in the resent, many researches shown that the β″ phase instead of β is the main strengthening precipitates in the Al-Mg-Si alloy, the stoichiometric composition of which supposed to be Mg5Si6 or Mg4Al3Si4, and the Mg/Si ratio (wt.%) is about 0.94 or 1.01. Our definition of “balanced” according to the Mg/Si ratio of β″ phase. But as being pointed out by the reviewer, such definition may lead to misunderstanding. We deleted the definition of “balanced alloy” in the manuscript, and revised the definition of the Mg/Si ratio used the value of Mg/Si ratio like 1.68, 1.00 and 0.58 Mg/Si ratio instead of high, low and balanced Mg/Si ratio. In section 2 we changed the description of the samples (Line 65-69): The Mg/Si ratio of the H-M and H-M-Sn alloy were 1.68 and 1.69 respectively, which were close to that of the equilibrium precipitates β; the Mg/Si ratio of the B-M and B-M-Sn alloy were 1.01 and 0.99 respectively, which were close to that of the main strengthening precipitates β″; the Mg/Si ratio of the L-M and L-M-Sn alloy were 0.58, which means that there were excessive Si according to the β″precipitates. The amount of Sn added is set 0.1% according to the investigation of Tu et al [20].
Advise 2: I am missing a direct comparison of the material properties you measured with results from other authors or commercial material (may also be from a data sheet of a company, cite it well). I propose to add a Strength/Elongation plot with your data points and add further data from other researchers or data sheets. If there is an offset of your properties in comparison to other works, this helps with the interpretation.
Response: Thank you for good suggestion. As the composition of investigated alloys is different to other researcher’s experimental alloys and commercial alloys, it is difficult to compare the mechanical properties of investigated alloys with results from other authors or commercial material. We have added the Strength/Elongation plot in the Figure 2 (d), which helps with the interpretation of the mechanical properties of the investigated alloys.
Reviewer 2 Report
In the introduction, the hypothesis should be mentioned in a sperate paragraph at the end with detailed work that authors have done. Authors in the introduction, when referring, written the full name of the first author of the articles they are citing. I would prefer to use it like Liu et al.; like a surname, then et al. Reference representation should be symmetrical throughout the text.
Author Response
Comment 2:
Advise 1: In the introduction, the hypothesis should be mentioned in a sperate paragraph at the end with detailed work that authors have done.
Response: Thanks for the reviewer’s insightful comment. We have detailed our previous work about addition of Sn in the Al-Mg-Si alloy in the end of Introduction by adding. (Line 53-55): Tu et al. found that the adverse effect of Sn on the artificial aging behavior of natural aged Al-Mg-Si alloy could be overcome by increase artificial aging temperature [20] or joint addition of Sn and Cu [21].
Advise 2: Authors in the introduction, when referring, written the full name of the first author of the articles they are citing. I would prefer to use it like Liu et al.; like a surname, then et al. Reference representation should be symmetrical throughout the text.
Response: We have revised the name of the refer authors in the manuscript, use it like Liu et al. (line 47) Shishido et al. and Cheng et al.; line 118 Tu et al. and Liu et al. [20, 33], (line …)
Reviewer 3 Report
It seems to me that the submission is of a good standard, with adequate English, and nicely reported. The experiment and its results are straightforward and are presented clearly.
Author Response
Comment 3:
Advise 1: It seems to me that the submission is of a good standard, with adequate English, and nicely reported. The experiment and its results are straightforward and are presented clearly.
Response: Thanks for the your kind suggestion and We will try our best to modify the manuscript. Best wishes for you.
Reviewer 4 Report
The present manuscript entitled “Effect of Trace Addition of Sn on Aging Behavior of Al-Mg-Si Alloy with Different Mg/Si Ratios” is interesting. However, authors need to justify the novelty as same authors have already published same work at https://doi.org/10.1016/j.msea.2019.138515 where they have added Cu and Sn both while in present study they have considered only Sn. It shows 27% similarity index. My following suggestion can improve the quality of the manuscript:
In title authors have mentioned “Ratios”. I think it should be “Ratio” instead of “Ratios”. The Abstract of the manuscript is unclear. It should be re written to brief clearly. In lines 15-18: the meaning about addition of Sn is identical as they mentioned that Sn addition significantly improve the age hardening and peak strength then there is no need to again mention that low Mg/Si addition with Sn reduce the hardenability and age strength. The state of the art in the manuscript have to be more concise and detailed. Authors need to provide reference in introduction section where they have mentioned that Sn addition can reduce the number density of atomic cluster……of the alloy. Which type of furnace have been used for the alloy synthesis and at which temperature the synthesis has been done? Authors need to provide the details of annealing instrument. In Fig. No. 1(b), suddenly M-M and M-M-Sn alloys are appeared in place of B-M and B-M-Sn alloys. If these are the same alloys, a uniform coding/designation should be maintained. The tensile strength value in present study is lower than 0.04% addition of Sn as published earlier. It means higher amount i.e. 0.1% Sn is giving negative results then what is the benefit to add higher amount of Sn. Line No. 116 should be modified. Why the tensile plasticity in H-M-Sn alloy is decreased whereas for other Sn-added alloys the same is increased? Line No. 147 “Mg/Si ratio in both the Sn-free alloys and Sn-add”- It should be rewritten as Sn-added alloys. Line No. 216 and 217, two symbols are different to indicate heating rate. Author should explain why the Avrami exponent (n) value is considered as 3/2? Line No. 265, the word “ration” should be changed into ratio.
Author Response
Comment 4:
Advise 1: The present manuscript entitled “Effect of Trace Addition of Sn on Aging Behavior of Al-Mg-Si Alloy with Different Mg/Si Ratios” is interesting. However, authors need to justify the novelty as same authors have already published same work at https://doi.org/10.1016/j.msea.2019.138515 where they have added Cu and Sn both while in present study they have considered only Sn. It shows 27% similarity index.
Response: Thanks for the reviewer’s kind suggestion. Our previous research found that Sn and/or Cu would affect the natural and artificial aging of certain Al-Mg-Si alloy. In this submitted manuscript, we have investigated what about the effect of Sn on aging behavior of Al-Mg-Si alloy would happen by changing the Mg/Si ratio of the alloys. In order to make this clear, we have revised the last paragraph of section 1 (line 53-61): Tu et al found that the adverse effect of Sn on the artificial aging behavior of natural aged Al-Mg-Si alloy could be overcome by increase artificial aging temperature [20] or joint addition of Sn and Cu [21].
Werinos et al. found that the natural and artificial aging of the Sn-containing Al-Mg-Si alloy were substantially affected by the concentration of Mg, Si and Sn [25]. However, what role the ratio of Mg/Si would play on the effect of Sn is unclear at present. Therefore, the effect of adding 0.1% wt. Sn on the artificial aging behavior of alloy with three different Mg/Si ratios aged at 180°C was investigated in the present paper, which provides experimental data and literature references for the design of Al-Mg-Si alloy.”
Advise 2: In title authors have mentioned “Ratios”. I think it should be “Ratio” instead of “Ratios”. In Fig. No. 1(b), suddenly M-M and M-M-Sn alloys are appeared in place of B-M and B-M-Sn alloys. Authors need to provide reference in introduction section where they have mentioned that Sn addition can reduce the number density of atomic cluster……of the alloy. If here are the same alloys, a uniform coding/designation should be maintained. Line No. 116 should be modified. Line No. 147 “Mg/Si ratio in both the Sn-free alloys and Sn-add”- It should be rewritten as Sn-added alloys. Line No. 216 and 217, two symbols are different to indicate heating rate. Line No. 265, the word “ration” should be changed into ratio.
Response: We have corrected the mistakes and revised the manuscript, like “Ratio” instead of “Ratios” and “M-M alloy” instead of “B-M alloy.” (line 3 and figure (b); line 118 (previous line 116) changed into “According to the investigation of Tu et al. [20] and Liu et al. [33]; All “Sn-add” and “Sn-add alloys” have been changed into “Sn-added alloys”; the in line 239 has been changed into .
Advise 3: The Abstract of the manuscript is unclear. It should be re written to brief clearly. In lines 15-18: the meaning about addition of Sn is identical as they mentioned that Sn addition significantly improve the age hardening and peak strength then there is no need to again mention that low Mg/Si addition with Sn reduce the hardenability and age strength.
Response: We have revised the Abstract by changing line 18-21: The results shown that Sn reduces the precipitation activation energy, increases the number density of β″ precipitates and then increased the ageing hardenability and mechanical properties of the Al-Mg-Si alloy. However, the positive effect of Sn on the mechanical properties of the Al-Mg-Si alloy drops with the decrease of the Mg/Si ratio of the alloy.
Advise 4: The state of the art in the manuscript have to be more concise and detailed. Which type of furnace have been used for the alloy synthesis and at which temperature the synthesis has been done?
Response: We have detailed the process of the casting experimental and list the model of the experimental instrument by changing line 71-73 into “Pure aluminum, pure magnesium and master alloys (Al-30%wt.Si, Al-10%wt.Sn and Al-10%wt.Ti) were melted in a graphite crucible with a resistance furnace (SG 2-7.5-10XP) at 750°C. The molten metals were casted into a copper mold after being degassed with C2Cl6 at 720°C.” and change line 78 into “An air circulating furnace (SGMA Z4/10A) was used for the above mentioned manufacture process.”
Advise 5: The tensile strength value in present study is lower than 0.04% addition of Sn as published earlier. It means higher amount i.e. 0.1% Sn is giving negative results then what is the benefit to add higher amount of Sn.
Response: Thanks for the reviewer’s kind suggestion. It is not surprise that the strength of the alloys with 0.1Sn are lower than that of the alloys in another paper (10.1016/j.msea.2019.138515) of us as Cu, Mn and Cr, which were supposed to increase the strength of Al-Mg-Si alloys, were added into the alloys investigated in that published paper. 0.1% instead of 0.04% Sn was added in the submitted manuscript was based on the result of our previous research (published paper with doi: 10.1016/j.msea.2019.138250) that the ability of decreasing the negative effect of nature aging can be increased by increasing the amount of added Sn from 0.05% to 0.1%.
Advise 6: Why the tensile plasticity in H-M-Sn alloy is decreased whereas for other Sn-added alloys the same is increased?
Response: Thanks for the reviewer’s kind suggestion. We have supplement illustrate of the reason of the different effects of Sn on the tensile plasticity of the investigated alloys in the part of Result in the manuscript. (Line 173-180): The length and diameter of the β'' precipitates of alloys with 1.00 and 0.58 Mg/Si ratio decreased substantially by the addition of Sn, and the smaller β'' precipitates have positive effect on elongation rate of alloy. The elongation rate of alloy with 1.00 and 0.58 Mg/Si ratio was increased by the addition of Sn, as shown in Figure 2 (d) and Table 2. However, the density of β'' precipitates increased significantly without not significantly decreased of the length and diameter of the β'' precipitates in the alloy with 1.68 Mg/Si ratio. That have negative effect on elongation ratio of alloy. The elongation ratio of alloy with 1.68 Mg/Si ratio was decreased by the addition of Sn, as shown in Figure 2 (d) and Table 2.
Advise 7: Author should explain why the Avrami exponent (n) value is considered as 3/2?
Response: We have provided the references to explain the reason of why we chose the value of the Avrami exponent (n) is 1.5. (Line 231-233: a value of 1.5 was found to be the optimal value of the Avrami index (n) for a description of the DSC curves analyzed [21, 29].)
Round 2
Reviewer 1 Report
Dear authors,
thank you for submitting your revised manuscript and you detailed explanations. I found no more issues.
Reviewer 4 Report
Authors have provided all details. Therefore, I accept it in present form.